# *Cuscuta* seeds: Diversity and evolution, value for systematics/identification and exploration of allometric relationships

**Magdalena Olszewski[1], Meghan Dilliott[1], Ignacio García-Ruiz[2], Behrang Bendarvandi[1], Mihai Costea[1] ***

**1** Department of Biology, Wilfrid Laurier University, Waterloo, Ontario, Canada, **2** Instituto Politécnico Nacional (CIIDIR-IPN Michoacán), Jiquilpan, Michoacán, México

* mcostea@wlu.ca

**Data Availability Statement:** All relevant data are within the manuscript and its Supporting Information files.

## Abstract

*Cuscuta* (dodders) is a group of parasitic plants with tremendous economic and ecological significance. Their seeds are often described as "simple" or "unspecialized" because they do not exhibit any classical dispersal syndrome traits. Previous studies of seed morphology and/or anatomy were conducted on relatively few species. We expanded research to 101 species; reconstructed ancestral character states; investigated correlations among seed characters and explored allometric relationships with breeding systems, the size of geographical distribution of species in North America, as well as the survival of seedlings. Seed morphological and anatomical characters permit the separation of subgenera, but not of sections. Identification of *Cuscuta* species using seed characteristics is difficult but not impossible if their geographical origin is known. Seeds of subg. *Monogynella* species, exhibit the likely ancestral epidermis type consisting of elongated and interlocked cells, which are morphologically invariant, uninfluenced by dryness/wetness. Subgenera *Cuscuta*, *Pachystigma* and *Grammica* have evolved a seed epidermis with isodiametric cells that can alternate their morphology between two states: pitted when seeds are dry, and papillose after seed imbibition. A seed coat with double palisade architecture throughout the entire seed has also apparently evolved in subgenera *Cuscuta*, *Pachystigma* and *Grammica*, but several species in two clades of the latter subgenus reverted to a single palisade layer outside the hilum area. The same latter species also evolved a peculiar, globose embryo, likely having a storage role, in contrast to the ancestral filiform and coiled embryo present throughout the remainder of the genus. Autogamous species had on average the highest number of seeds per capsule, whereas fully xenogamous taxa had the lowest. No correlation was revealed between the size of the seeds and the size of their geographical distribution in North America, but seedlings of species with larger seeds survived significantly longer than seedlings resulted from smaller seeds. Diversity and evolution of seed traits was discussed in relationship with their putative roles in dormancy, germination and dispersal.

**Funding:** MC lab was funded by NSERC Discovery Canada (213918): https://www.nserc-crsng.gc.ca/professors-professeurs/grants-subs/dgigp-psigp_eng.asp The funder had no role in study design, data collection and analysis, decision to publish, or preparation of the manuscript.

**Competing interests:** The authors have declared that no competing interests exist.

## Introduction

*Cuscuta* (Convolvulaceae; dodders) is a genus of about 200 species of obligate parasitic plants with sub-cosmopolitan distribution (reviewed by [1]). Dodders are keystone species in their natural ecosystems impacting the diversity, structure and dynamics of plant communities [e.g., 2]. *Cuscuta* is one of the most economically detrimental groups of parasitic plants worldwide as infestation by some of its species can result in major yield losses in numerous crops [3–7].

Seeds are either dispersed or persistent in a seed bank [6–8], and thus important both from applied and theoretical points of view. The worldwide anthropogenic dispersal of *Cuscuta* seeds through contaminated commercial seed shipments and herbal products has been well documented [e.g., 4, 6, 9]. The seeds of *Cuscuta* lack apparent morphological adaptations for a particular dispersal syndrome and have been considered "simple" or "unspecialized" [e.g., 10–12]. However, dodder seeds have been recently reported to be long-distance dispersed via bird endozoochory [13, 14] or water in the species with indehiscent fruits [15].

In an effort to prevent and mitigate the threat of *Cuscuta* as invasive plants and agricultural weeds, quarantine legislation has been enacted worldwide [e.g., 6, 16]. Enforcing such legislation internationally is predicated on the ability to identify *Cuscuta* seeds, and morphological identification has remained prevalent in many phytosanitary labs worldwide because it is more expedient than the molecular approaches. Unfortunately, the taxonomy of *Cuscuta* species has historically relied on flowers and to a less extent on fruits [1, 15, 17, 18]. To date, the seeds of only 22 *Cuscuta* species have been studied; usually a few species at a time and often either morphologically or anatomically (summarized in S1 Table; e.g., 9, 12, 19–24). Therefore, an overarching study of *Cuscuta* seeds with a broader taxonomic sampling is necessary to unify previous results, as well as to provide a comprehensive source of data for the comparison of seed characters with identification potential among species. Surveying the morphological and anatomical diversity of seeds in a phylogenetic framework (e.g., tracing character evolution) would also be important for the systematics of *Cuscuta* because of the scarcity of available morphological characters that bear a phylogenetic signal in this genus [1, 10, 25].

The dataset of seed traits resulted from this study can also be used to explore possible allometric or functional relationships of seeds in *Cuscuta*; for example, the average number of seeds produced per fruit and the breeding systems; the size of seeds versus the geographical distribution range of species, as well as seed size and the survival of seedlings. Broad-scale comparative studies of *Cuscuta* pollen/ovule ratios indicated that dodders possess a wide range a mixed mating systems, which ranged from functionally cleistogamous (and thus selfing) to obligate xenogamous [26, 27]. While the number of ovules per ovary is always four, pollen production by each flower varies over three orders of magnitude [27]. This allows testing of possible relationship between pollen ovule/ratios and the average number of seeds produced per capsule [e.g., 28–30]. Seed size has been related in other angiosperms to dispersal [e.g., 31–33] and seedling survival [34, 35]. This latter aspect is particularly important for *Cuscuta* population dynamics because although their seedlings are capable to uptake water and even form short-term associations with mycorrhizal fungi [36], during this stage they rely entirely on the nutritive reserves stored in the endosperm. If seedlings cannot locate and attach to a compatible host within the short window of time provided by the seed reserves, they will die [4, 6]. Anecdotal *Cuscuta* seedling survival times reported in the literature (i.e., which did not result from a study) vary from eight days (*C. campestris*; [3]) to seven weeks (*C. gronovii*; [37, 38]), but to date there has been no study comparing the seedling survival of several species of *Cuscuta*.

Thus, the objectives of this study are: (a) Survey the morphological and anatomical diversity of *Cuscuta* seeds, reconstruct ancestral character states and investigate correlations among characters; discuss the usefulness of seed characters for species identification and their significance to the taxonomy and systematics of the genus; (b) Examine a putative relationship between the number of seeds per capsules and breeding systems in *Cuscuta*; (c) Establish if there is a correlation between seed size and the distribution range size of species in North America; (d) Experimentally determine the effect of seed size on the seedling survival in three *Cuscuta* species.

## Materials and methods

### Taxonomic sampling

The morphology and anatomy of seeds was examined in 101 *Cuscuta* species representing all currently accepted *Cuscuta* subgenera and sections, except the small sect. *Epistigma* of subg. *Cuscuta* [1] (S1 Appendix). All the seeds were obtained from herbarium specimens and two to six specimens were examined per taxon (S1 Appendix). Specimens from the following herbaria were annotated and sampled: AAU, ALTA, ARIZ, ASU, BRIT, CANB, CAS, CHSC, CICY, CIMI, CTES, DAO, F, G, GH, IBUG, IEB, IND, JEPS, K, LL, MEL, MICH, MEXU, MO, NBG, NMC, NMS, NY, OSC, OSU, QCNE, QFA, QUE, RSA, S, SD, SMU, TEX, TRTE, UBC, UC, UCR, UNB, UPS, UPRRP, USAS and WLU (abbreviations from Index Herbariorum; S1 Appendix).

### Seed morphology and anatomy

Ten seeds per herbarium specimen were used for external morphology using Scanning Electron microscopy (SEM). Seeds were rehydrated in a 50% ethanol solution brought to boiling point and preliminarily examined/imaged with a Nikon SMZ1500 stereomicroscope. Seeds were dehydrated through an ethanol series (50%, 70%, 85%, 95%, and 100%; each step 1h) and then dried with a Tousimis Autosamdri-931 critical point dryer. Using SEM has been proven to be a valid approach to study not only the external morphology of seeds, but also the seed coat anatomy in *Cuscuta* [13, 14]. For anatomy, 5–10 seeds were sectioned longitudinally by hand with a razor blade along the hilum area. Longitudinal sections were dehydrated as for the SEM processing. Critical-point dried, entire and longitudinally sectioned seeds were sputter-coated with 30 nm of gold using an Emitech K550 (Emitech, Ltd. Ashfort, UK). Imaging was completed using a Hitachi SU-510 variable pressure scanning electron microscope (SEM) at three kV. Character scoring and measurements were done using Quartz PCI version 5.1 (Quartz Imaging Corp.).

In addition, five seeds of 27 species representing the three main subgenera (subg. *Monogynella*: *C. exaltata*, *C. gigantea*, *C. monogyna*, and *C. reflexa*; Subg. *Grammica*: *C. californica*, *C. campestris*, *C. argentiniana*, *C. cephalanthi*, *C. acutiloba*, *C. chapalana*, *C. chilensis*, *C. compacta*, *C. corymbosa*, *C. cristata*, *C. cuspidata*, *C. denticulata*, *C. desmouliniana*, *C. erosa*, *C. foetida*, *C. grandiflora*, *C. iguanella*, *C. indecora*, *C. mitriformis*, *C. veatchii*; subg. *Cuscuta*: *C. epilinum*, *C. epithymum*, *C. europaea*) were also cross-sectioned by hand and processed for optical microscopy. These latter sections were stained with 0.05% Toluidine Blue O (TBO), a polymorphic stain (pH 4.4; [39]); Sudan IV for lipids [40], and potassium iodide for starch ($I^2KI$; [39]).

Seed water gap was studied in seven species from three of the four subgenera (Subg. *Monogynella*: *C. monogyna*, *C. lupuliformis*; Subg. *Grammica*: *C. gronovii*, *C. sandwichiana*, *C. tasmanica*, *C. veatchii*, *C. volcanica*; Subg. *Cuscuta*: *C. epithymum*) using the protocol developed by Jayasuiraya et al. [22]. Twenty seeds per species were processed. Physical dormancy was

removed from 14 of the 20 seeds using the rehydration protocol mentioned above. Non-dormant seeds can be recognized by the open hilar fissure [22]. The hilum region of seven of the non-dormant seeds was painted with petroleum gel to obturate the hilar fissure, while in the other seven seeds, the hilum fissure was left open. The remaining six dormant seeds were not treated in any way to serve as a control. Both dormant and non-dormant seeds were placed in an aqueous solution of 25% Aniline Blue, in glass trays with one seed per basin. Seeds were removed at 15 min intervals from the solution. After 15 min to 2h, seeds were longitudinally sectioned by hand through the hilar pad, along the hilar fissure, to observe the penetration of dye. Observation and imaging of the cross-sections and water gap samples was conducted using a Nikon SMZ1500 stereo-microscope and imaged with a PaxCam Arc digital camera equipped with Pax-it! 2 Version 1.5 software (MIS Inc, Villa Park, IL).

### Ancestral reconstruction and correlations among seed traits

Nine categorical and 13 continuous characters (Table 1) were generated based on the available *Cuscuta* seed morphological and anatomical literature (S1 Table). Description of shapes was based on [41]. Three additional characters consisting of ratios between anatomical continuous features and the seed length (Table 1) were added after the initial character scoring. Basic statistics (e.g., averages, standard deviations, normal distribution tests) and Pearson's correlations were conducted using PAST version 3.16 [42].

Character states were mapped onto a recent genus phylogeny based on *rbcL* and nrLSU [43]. Distribution of characters was analyzed only in-group as the position of *Cuscuta* within Convolvulaceae is currently not resolved [44]. Scenarios of character evolution were analyzed using the parsimony reconstruction method provided by Mesquite 3.40 [45]. Markov k-1 state 1 parameter model (MK1) of evolution was used. In the parsimony reconstruction, character-state changes were treated as unordered. Three qualitative, non-polymorphic characters (outer palisade layer presence, epidermal cell type and type of embryo) were also analyzed with the likelihood reconstruction method [45]. The correlation between the seed epidermal shape and their ability to reverse between pitted and papillate (binary characters, Table 1) was determined using Pagel's method [46] implemented in Mesquite.

### Number of seed per capsule and breeding systems

We used the pollen/ovule (P/O) ratio data published by Wright et al. [27]. The latter authors had also assigned taxa to breeding system categories based on Cruden's ranges [47]: six species were inferred to be fully xenogamous, 108 taxa facultatively xenogamous and at least 23 taxa facultatively autogamous [27]. Differences among P/Os and number of seeds per capsule (S/C) averages were analyzed using an Analysis of Variance (ANOVA). Additionally, a regression tree was constructed ("r.part"–[48]). The defined response variable was the P/O ratio, the explanatory variables the breeding system categories, and the average number of seeds per capsule the prediction model.

### Seed size and distribution range of *Cuscuta* species in North America

Geographical distribution range size (km$^2$) data for 50 North American *Grammica* species were taken from Ho and Costea [15]. As seed length was strongly positively correlated with both the width and thickness of seeds (see Results), it was selected to represent the "seed size" variable. Geographical range size data did not follow a normal distribution [15], and a Spearman's Rank Correlation with seed size was conducted using PAST version 3.16 [42].

**Table 1. Seed characters surveyed and their representative codes and states.** Continuous characters values are averages.

| Character | Character states |
|---|---|
| **Categorical characters** | |
| 1. Compression of seed or the number of ± flat faces that a seed has. | 1 = dorsoventrally compressed are seeds with one flat face and one convex face; 2 = "angled", seeds with 2 flat faces and one convex face; 3 = no compression; spherical or ovoid |
| 2. Seed shape (the part with the hilum was considered the base of the seeds) | 1 = elliptic; 2 = obovate; 3 = circular; 4 = ovate, 5 = oblong |
| 3. Radicular end of embryo | 1 = spherically enlarged; 2 = filiform |
| 4. Hilum position | 1 = terminal; 2 = subterminal |
| 5. Hilum compression | 1 = flat; 2 = concave |
| 6. Dry seed epidermal cells | 1 = pitted; 2 = non-pitted |
| 7. Hydrated seed epidermal cells | 1 = papillose; 2 = non-papillose |
| 8. Seed epidermis cell shape (as seen in surface SEM images) | 1 = elongated; 2 = isodiametric |
| 9. Presence of outer palisade layer | 0 = absent; 1 = present |
| **Continuous characters** | |
| 10. Number of embryo coils (a "coil" represents a 360˚ rotation of the filiform embryo) | - |
| 11. Number of seeds per capsule | - |
| 12. Seed length (usually the longest axis, measured from the hilum to the distal part of seeds) | μm |
| 13. Seed width (the axis perpendicular on the length and in the same plane with it) | μm |
| 14. Seed thickness (the axis perpendicular on the plane formed by the length and width). | μm |
| 15. Hilum area length | μm |
| 16. Hilum area width | μm |
| 17. Length of funicular scar of the hilum | μm |
| 18. Epidermal cell diameter (surface morphology) | μm |
| 19. Thickness of epidermal cell (anatomy) | μm |
| 20. Width of epidermal cell (anatomy) | μm |
| 21. Thickness of outer palisade layer (anatomy) | μm |
| 22. Thickness of inner or single palisade layer (anatomy) | μm |
| **Ratios** | |
| 23. Ratio of epidermal cell diameter and seed length. (Subg. Monogynella taxa were not included because of their different epidermal cell morphology). | - |
| 24. Ratio of epidermal cell thickness and seed length | - |
| 25. Ratio of inner + outer palisade thickness and seed length | - |

## Seed size, germination and seedling survival

A comparative seedling survival experiment was conducted in three species, *C. epithymum* (subg. *Cuscuta*), *C. costaricensis* and *C. campestris* (subg. *Grammica*), which have seeds of different sizes [49–51]. Herbarium collections sampled had been collected in 2007 (*C. campestris*) and 2018 (*C. epithymum* and *C. costaricensis*) (S1 Appendix). To corroborate the seed size differences, 500 seeds of each species were imaged and measured using a Nikon SMZ1500 stereomicroscope using Pax-it ver. 1.4.2.0 software and a PaxCam Arc digital camera (MIS Inc., Villa

Park, IL). Seed weight was also determined using a Cole-Parmer Symmetry PA - 124I analytical balance. Basic statistics, normality, and a one-way Analysis of Variance (ANOVA) were performed to verify that seed size was significantly different among the three species using PAST version 3.16 [42].

The physical dormancy of *C. campestris* and *C. costaricensis* seeds was broken by a scarification treatment in 99.99% sulfuric acid for 30 minutes, after which seeds were rinsed with sterile Milli-Q water, submerged in bleach for three minutes and then thoroughly rinsed again with sterile water. As the seeds of *C. epithymum* are known to possess a combinational physical and physiological dormancy [8], after the sulfuric acid stratification, they received an additional treatment of gibberellic acid (GA) 1000 ppm [52].

Treated seeds of each species were transferred into sterile, 140 mm sterile Petri dishes each with two Whatman filter paper rings moistened with 15 ml of sterile Milli-Q water. The Petri dishes with seeds were then incubated at 32°C and light (150 mmol $m^{-2}$ $s^{-1}$, 12h/day) for germination. Once the tip of the radicle-like organ emerged approximately 1 mm from the seed coat, seeds were considered germinated and were transferred to smaller sterile Petri dishes (90 mm) prepared with one Whatman filter paper ring and 5 ml sterile Milli-Q water. In total, 150 seedlings per species, distributed two per Petri dish, we examined. Petri dishes were sealed with Parafilm M and placed in the greenhouse at 18°C/21°C, 8/16 h (light intensity 39.6 μmol $m^{-2}$ $s^{-1}$) fully randomized. Seedlings were monitored daily and re-randomized every three days. Sterile Milli-Q water was added throughout the study to maintain the filter paper humid. Seedlings were considered "dead" once the entire seedling was necrotic, from the radicular end to shoot tip. Seedling survival data was analyzed using Kaplan-Meier survival curves with XLSTAT version 2019.4.2.

## Results

### External morphology and micromorphology; ancestral reconstruction and correlations

Characters surveyed are outlined in Table 1 and their complete scoring is presented in S2 Table.

Seeds of *Cuscuta* develop within a two-locular ovary, with constantly two anatropous, unitegmic ovules per locule. However, 1–4 mature seeds will develop per capsule (S2 Table). As seeds develop in close proximity to one another, the number of seeds per locule determines their compression morphology (Table 1). When seeds adjoin within the same locule, they will possess an "angled" morphology; one seed per locule leads to a dorsoventrally compressed morphology, and one seed per capsule will result in a non-compressed morphology, spherical to ovoid. As the number of seeds varied somewhat within each taxon from capsule to capsule, two or three compression character states were observed in about 65% of taxa (S2 Table). However, the average number of seeds developed per capsule (S/C) was relatively constant in each taxon and varied from 1 to 3.8 (S2 Table). Capsules with constantly one seed per fruit evolved in seven *Grammica* clades (S2 Table). In general, species of subg. *Cuscuta* averaged the highest S/C (3.1–3.8), and therefore a majority of their seeds were "angled". Hilum position is also associated with the compression morphology: a lateral hilum is present in species with dorsi-ventrally compressed seeds, while a terminal hilum was observed in "angled" and non-compressed seeds (S2 Table). The most common seed shapes encountered were elliptic and ovate, while oblong and circular seeds were observed less frequently; however, most species exhibited a combination of two or three seed shape character states (S2 Table).

Epidermis cell shape is always correlated with the ability of seeds to revert between pitted and papillose morphology (Pagel's test, 5000 simulations, p = 0) and two types of seed epidermises were distinguished:

1. Type I (Fig 1A) has rectangular, elongated epidermal cells; parallel groups of 2–6 such cells are perpendicular on the long axis of similar groups of cells. Epidermal cells are invariant morphologically, unaffected by dryness and wetness (see the next type). Type I characterizes species of subg. *Monogynella*.

2. Type II (Fig 1B–1F) has more or less isodiametric epidermis cells that can shift their morphology alternating between two states: either pitted (concave) when seeds are dry (Fig 1B–1D), or dome-shaped, papillose (convex) when seeds are hydrated (Fig 1E and 1F). This is the most common seed epidermis in dodders, present in the subgenera *Cuscuta*, *Pachystigma* and *Grammica*.

Ancestral parsimony reconstruction of the two types of epidermises was equivocal while maximum likelihood reconstruction weakly supported Type I as the ancestral character state (Proportional likelihood Type I: 0.5585; Type II: 0.4414; Fig 2A). Considering the putative evolutionary advantages provided by Type II for seed imbibition (see Discussions), this is indeed most likely the derived character state.

The micropyle is completely obstructed during seed development and no remnants of it were observed in the mature seeds. The hilar area is always a morphologically distinct region, regardless of the type of seed coat. Hilum area is round to elliptic and has in the center the funicular scar or hilar fissure (Fig 1G and 1H). Epidermal cells of the hilar pad are substantially smaller than in the rest of the seed epidermis; rectangular-elongated and concentrically arranged around the hilar fissure (Fig 1G and 1H). Length, width, and size values of hilar fissure were the highest in subg. *Monogynella*—in some species three to four times larger than in remaining subgenera (S2 Table). Although decreasing in size in subg. *Grammica*, species of Clades D (sect. *Oxycarpae*) and G (sect. *Lobostigmae*), had also relatively large hilar pads and fissures (S2 Table).

In general, there are several degrees of variation among taxa across the genus in regard to the quantitative characters (e.g., seed length, hilar pad size), however within each taxon, the variation was relatively consistent (S3 Table). For example, seed length ranged from 704.55 μm to 3158.30 μm, while width varied from 666.28 μm to 2910.5 μm (S3 Table). Seed size within species had a standard deviation of as little as 16.87 μm (*C. membranacea*) to as much as 196.2 μm (*C. monogyna*), indicating that seed size is a relatively reliable character within each species. Seeds of subg. *Monogynella* are the largest, whereas those of subg. *Cuscuta* are the smallest (S2 Table). Subgenus *Grammica* species exhibit the most extensive variation of seed size, for example taxa in Clade G (sect. *Lobostigmae*) had an average seed length of 1658 μm and width of 1420 μm, while taxa of Clade L (sect. *Umbellatae*) had a seed length of 972 μm and width of 846 μm (S2 Table). Seed length was strongly correlated with seed width and thickness (S4 Table). Similarly, the length of the hilum area was strongly correlated with its width and the length of the funicular scar (S4 Table).

When mapped into the genus phylogeny, all the quantitative characters exhibited extensive homoplasy. For example, subg. *Grammica* taxa of Clade D (sect. *Oxycarpae*) and Clade G (sect. *Lobostigmae*) have evolved similar seed length, epidermal cell thickness, inner and outer palisade thickness, whereas, taxa of Clade A (sect. *Californicae*) and Clade L (sect. *Umbellatae*) have similar hilar pad length and width and palisade layer(s) thickness.

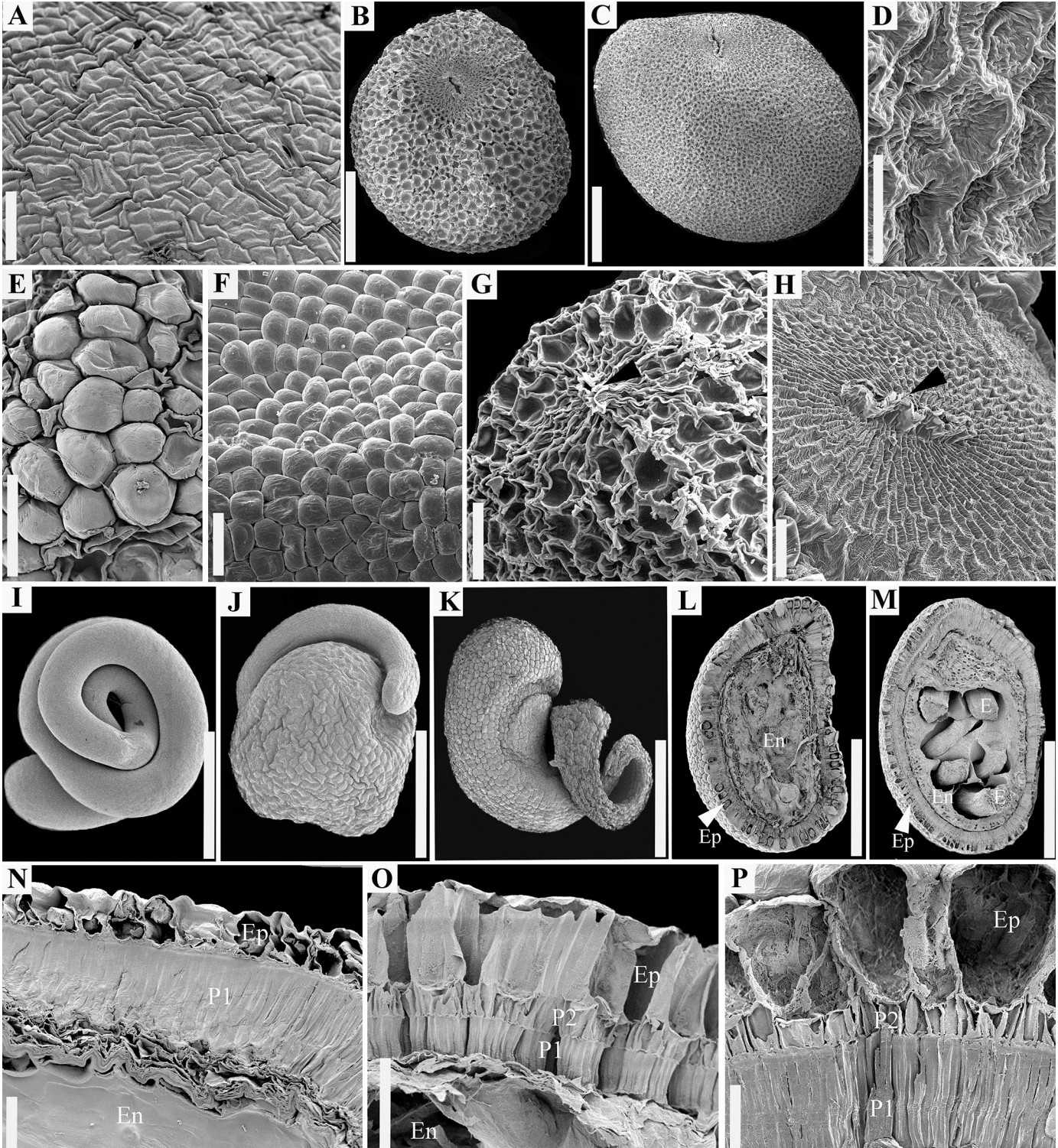

**Fig 1. Seed features revealed with scanning electron microscopy.** A. Epidermis surface, *C. lupuliformis* (subg. *Monogynella*). B–D. Overall seed morphology and surface of dry epidermis. B. *Cuscuta europaea* (subg. *Cuscuta*). C–D. *Cuscuta cephalanthi* (subg. *Grammica*). C. Entire seed. D. Detail of pitted epidermis of dry seeds. E–F. Different stages of epidermis rehydration. E. *Cuscuta gronovii* var. *gronovii*. F. *Cuscuta cephalanthi*. G–H. Hilum area (black arrows indicate hilar fissure). G. *Cuscuta approximata*. H. *Cuscuta mitriformis*. I–K. Embryo morphology. I. Filiform and coiled, *C. pacifica*. J–K. Globose toward the radicular end. J. *Cuscuta nevadensis*. K. *Cuscuta microstyla*. L–M. Longitudinal sections through the hilum area showing all the seed components. L. *Cuscuta epithymum*. M. *Cuscuta globulosa*. N–P. Seed coat anatomy. N. *Cuscuta lupuliformis*. O. *Cuscuta alata*. P. *Cuscuta gronovii* var. *gronovii*. Ep = epidermis; En = endosperm;

E = Embryo; P1 = Inner or single palisade layer; P2 = Outer palisade layer. Scale bars. A, E, F = 200 µm; D = 40 µm; G, H = 100 µm; B, C, I–M = 0.5 mm; N–P = 50 µm.

### Anatomy of seed coat; ancestral reconstruction and correlations

The seed coat originates from the single ovule integument and has a simple structure, consisting of a two or three cell layered testa and several crushed parenchymatic cells representing the tegmen. The embryo is filiform, coiled within the endosperm.

Seen in longitudinal sections, Type I epidermal cells (of subg. *Monogynella*) appear more or less rectangular and contain abundant tannins. Type II epidermal cells (of subgenera *Grammica*, *Cuscuta* and *Pachystigma*) are radially elongated, tapered basally and rounded distally. Developing seeds had starch grains in their epidermis cells. The ratio between the epidermal cell thickness and the seed size was highest in subg. *Cuscuta* and smallest in subg. *Monogynella* (S2 Table).

The majority of *Cuscuta* species possess an inner and an outer palisade layer, which are continuous ("complete") throughout the entire seed coat, including in the hilar area (Fig 1L and

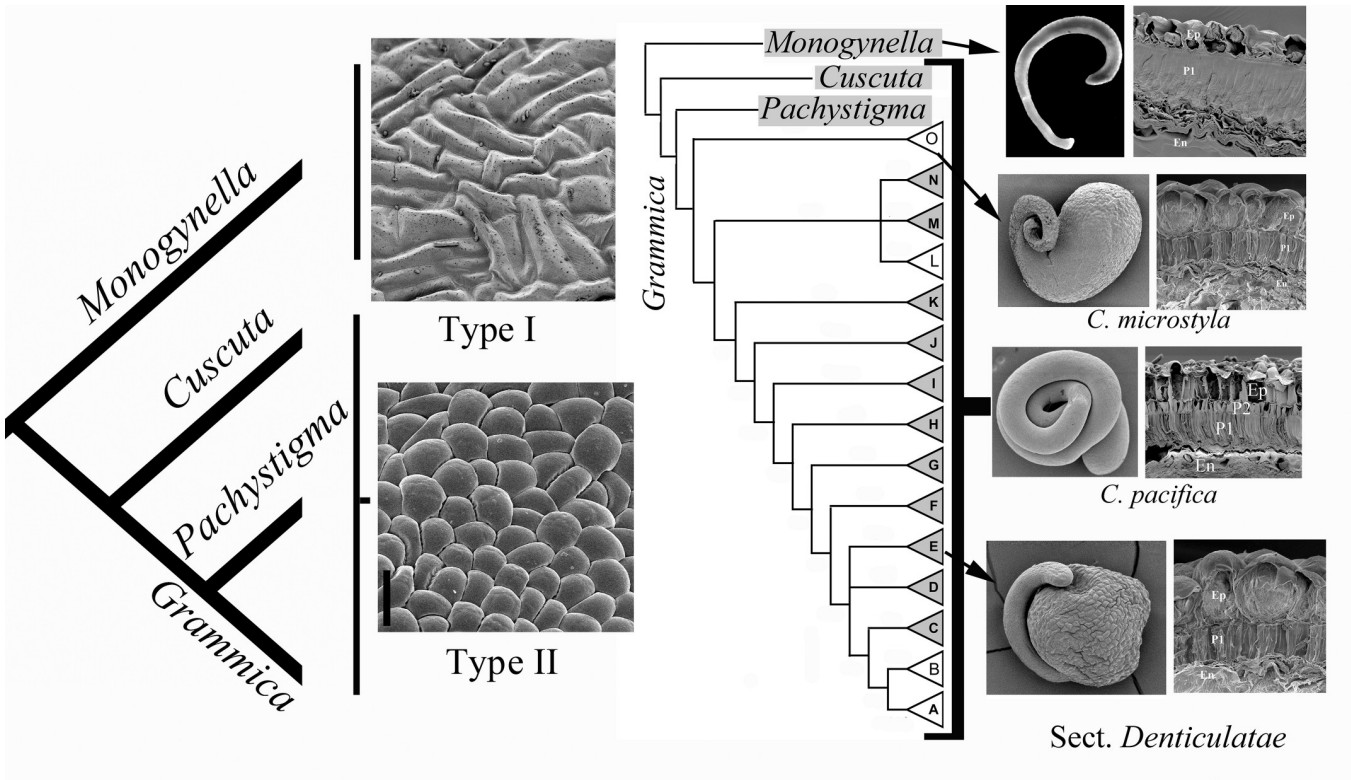

## A. Epidermis morphology          B. Embryo and architecture of palisade layers

**Fig 2. Summary of character evolution hypotheses.** A. Evolution of epidermis morphology. An invariant epidermis with rectangular, interlocked epidermal cells (Type I) is likely ancestral and characterizes subg. *Monogynella*. An epidermis with isodiametric cells that can alternate their morphology between dome-shaped and pitted (Type II) evolved in subgenera *Cuscuta*, *Pachystigma* and *Grammica*. B. Evolution of embryo and architecture of palisade layers outside hilum area. The seed coat with only one palisade layer (P1) outside hilum area in subg. *Monogynella* is likely ancestral, while a seed coat with two palisade layers (P1 and P2) in the remaining subgenera, is likely derived; one palisade layer reverted two times in subg. *Grammica* in *C. microstyla* (clade O) and sect. *Denticulatae* (clade E). The latter taxa also evolved an embryo with an enlarged radicular end, which likely functions as a storage organ. Ep = epidermis; En = endosperm.

1M). As an exception, in subg. *Monogynella* and four species of subg. *Grammica* (see below), the outer palisade layer is "incomplete", present only in the hilar region, and absent from the rest of the seed coat where only a single palisade layer can be observed (Figs 1N, 2B and 3E–3H). The double palisade structure originates from a periclinal division of the same cell layer that does not divide in the case of the single palisade layer architecture. Palisade layer(s) of immature seeds possess thin cellulosic cell walls and contain abundant starch grains (Fig 3I). The inner and single palisade layers cells undergo a secondary thickening through deposition of lignin, which obturates their lumen almost entirely (Figs 1N–1P, 3H, 3J and 3K). Thus, the inner palisade layer (in case of the double palisade structure) and the single palisade layer (for the single architecture) are homologous. These latter palisade layer cells exhibit a *linea lucida* (or "light line"), a light refractive, apparently denser region in the upper third of radial cell walls (Fig 3H, 3J and 3K). The outer palisade layer (in the case of the double palisade architecture outside the hilum) cells are shorter than those of the inner palisade layer; they do not exhibit a light line and their cells walls remain relatively thin, although also lignified (Fig 1O and 1P). The thickness of the external and internal palisade layers is strongly correlated (S4 Table). Also, a moderate positive correlation was retrieved between the thickness of the external palisade layer and of the epidermis cells (S4 Table). As palisade layer(s) represent(s) the mechanical layers, based on the classification of [53], it results that *Cuscuta* seeds are endotestal.

Likelihood reconstruction weakly supported a derived status for the double palisade layer from the single palisade architecture (proportional likelihood = 0.5586; Fig 2B). Reversals to a single palisade layer anatomy have occurred two times in four species of two clades within subg. *Grammica*: all the species of sect. *Denticulatae* (Clade E; *C. denticulata*, *C. nevadensis* and *C. veatchii*) and *C. microstyla* in sect. *Subulatae* (Clade O) (Fig 2B).

The hilar pad epidermis cells are small, rectangular and thin-walled, cellulosic. As indicated above, the seed coat structure within the hilar pad is invariant across the entire genus being always composed of two palisade layers. Especially the inner palisade layer increases significantly in thickness (up two times) in the hilum area compared to its size in the rest of the seed coat. A suture-type discontinuity within the epidermis and palisade layers at the centre of the hilar pad forms the hilar fissure (Fig 3E and 3F). This is also where the seed water gap is located. Tracheids, which are most likely remnants of the funiculus vasculature, were observed in this region, embedded in a parenchymatic tissue that is not part of the endosperm (Fig 3F and 3G). The dye tracking experiment revealed that although the epidermis cells of dormant seeds hydrated (which can be determined by their bulging and absorbing of stain), the aniline blue solution did not penetrate through the palisade layer(s) even 60 min after soaking in the dye. In contrast, in non-dormant seeds, the dye began to infiltrate through the hilar fissure (including through the parenchyma in the area) after 15 min (Fig 3M). After 120 min, the stain was observed around the endosperm and embryo of non-dormant seeds. The dye also infiltrated into the endosperm and embryo via irregular fissures within the palisade layers caused by accidental mechanical injury during processing.

The incipiently developing endosperm is nuclear and many free nuclei were observed; however, eventually cell wall formation is initiated and gradually progresses centripetally. Endosperm of young seeds is starchy and becomes "gelatinous" in mature seeds. Gelatinization is apparent only in hydrated seeds; dry seeds have a "hard" endosperm. A peripheral, "membranous", cell layer with large nuclei (called aleurone layer by [12, 20]) was observed around the endosperm, demarcating it from the parenchymal layers of the tegmen. Ten species of subgenera *Grammica* and *Pachystigma* (*Grammica*: *C. sandwichiana*—Clade B (sect. *Racemosae)*; *C. nevadensis*, *C. denticulata*, *C. veatchii*—Clade E (sect. *Denticulatae*); *C. haughtii*–Clade F (sect. *Partitae*); *C. tinctoria*–Clade G (sect. *Lobostigmae*); *C. strobilacea*–Clade K (sect.

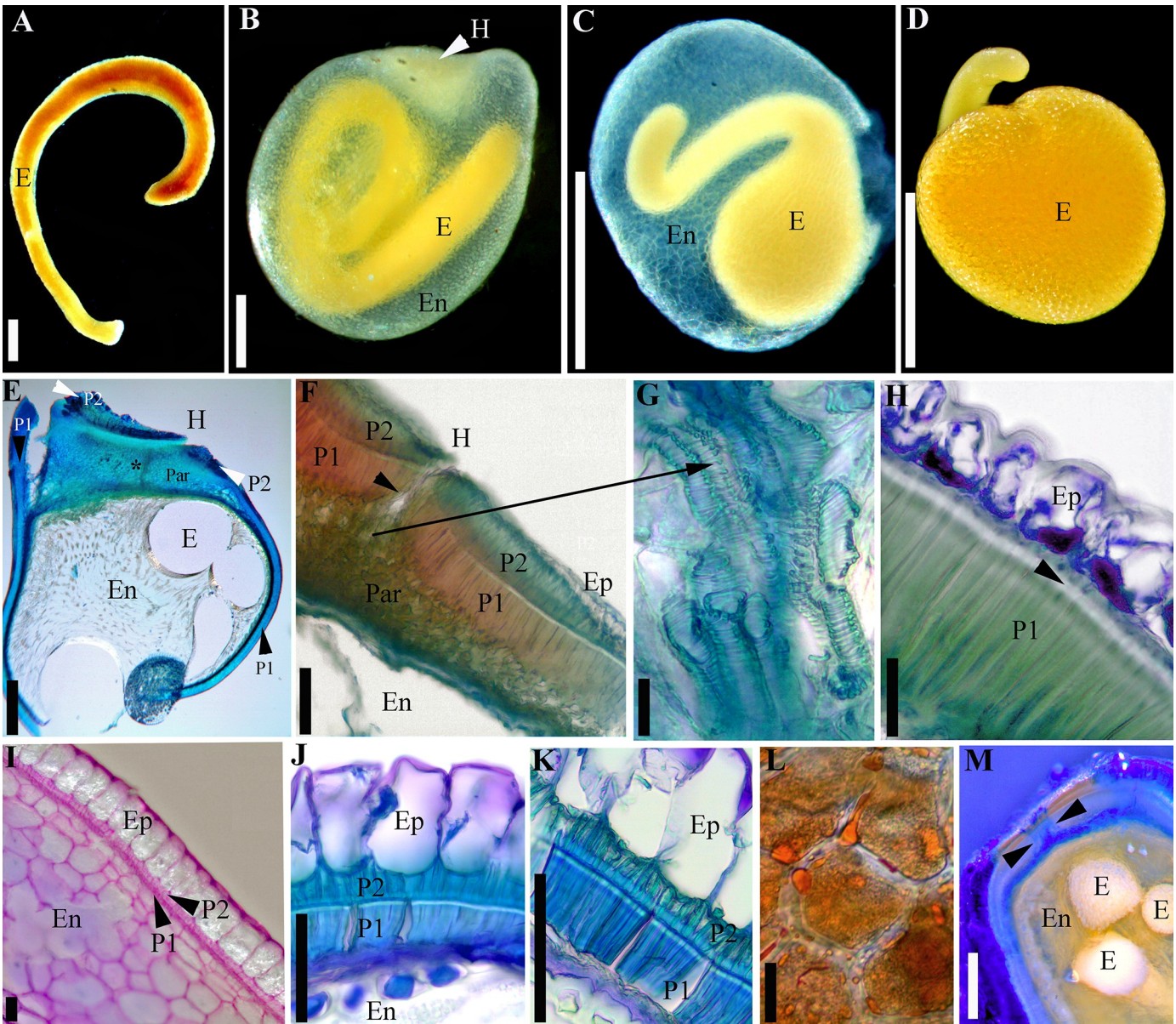

**Fig 3. Seed features viewed with light microscopy.** A–D. Embryos. A. *Cuscuta monogyna* (embryo removed from the endosperm); note that in this species the embryo does not form a full coil. B. Embryo of *C. tinctoria* var. *floribunda* embedded in the endosperm. C. Developing embryo of *C. nevadensis* surrounded by the endosperm epidermis (the rest of endosperm was almost entirely consumed). D. Fully developed embryo of *C. nevadensis* (endosperm epidermis removed). E–H. *Cuscuta lupuliformis* (subg. *Monogynella*). E–G. Longitudinal sections through the hilum area of *C. lupuliformis*. E. Overview; black asterisk indicates position of water gap. F. Detail of hilum area; black arrow indicates position of water gap; note that two palisade layers (P1 and P2) are present. G. Tracheid-like structures embedded in a parenchyma tissue. H. Testa architecture outside hilum area with only one palisade layer; black arrow indicates *linea lucida*. I–K. Seed coat architecture with two palisade layers outside the hilum area. I. Incipient stage in the development of the two palisade layers in *C. argentinana*; at this stage, epidermis contains starch grains. J. *Cuscuta europaea*. K. *Cuscuta cristata*; note the presence of *linea lucida* in the inner palisade layer (P1). L. Parenchyma cells with lipids and starch in the enlarged portion of *C. nevadensis* embryo. M. Longitudinal section of rehydrated *C. sandwichiana* seed after 15 min in Aniline Blue; dye penetration is limited to the water gap (indicated with arrows). E = Embryo; En = endosperm; H = hilum; Ep = epidermis; Par = parenchyma; P1 = Inner or single palisade layer; P2 = Outer palisade layer. Scale bars. A–E = 0.5 mm; G, I–K = 50 µm; H, L = 25 µm; F, M = 100 µm.

*Ceratophorae*); *C. acuta*–Clade L (sect. *Umbellatae*); *C. microstyla*–Clade O (sect. *Subulatae*); *Pachystigma*: *C. nitida*) displayed a markedly thicker endosperm epidermis, which separated easily from the rest of the seed coat.

The embryo is devoid of meristems at the radicular end, and has no cotyledons. The number of embryo coils varies; in some species it appears to curve resembling a cane or a hook, but no coils form (Fig 3A), while in others the embryo displays anywhere from one to four and half coils (Figs 1A, 3B and 3C; S2 Table). The number of coils varies considerably amongst subgenera, but remains relatively consistent within each species. Subgenera *Monogynella*, *Pachystigma* and *Cuscuta* possess the lowest number of coils (1–2), while an increased number of coils has evolved multiple times in subg. *Grammica* (S1 Fig). A peculiar embryo, spherically enlarged toward the radicular end has evolved in four species of subg. *Grammica*: *C. denticulata*, *C. veatchii*, *C. nevadensis* (Clade E–sect. *Denticulatae*; Figs 1J, 3C and 3D) and in *C. microstyla* (Clade O–sect. *Subulatae*; Fig 1K). These taxa exhibit a spherical swelling toward the radicular end of the embryo, differing only in the size of the globose part and the number of coils in the distal, filiform part toward the shoot (between zero and 1.5 coils). The endosperm of these taxa is much reduced compared to the other *Cuscuta* species, limited to several marginal cell layers which are entirely consumed during embryo development in such a way that when seeds are mature, the endosperm is represented only by its thick epidermis. The globose radicular part consists of an epidermis and a storage parenchyma with starch and lipid droplets (Fig 3L). It should be noted that this remarkable embryo morphology is associated in these species with the reduction of the testa to a single palisade layer outside the hilum area (Fig 2B). This type of embryo has clearly evolved from the ancestral filiform embryo characteristic for the remainder of the genus (Fig 2B).

## Breeding systems and number of seeds per capsule in *Cuscuta*

ANOVA indicated a significant relationship between the number of seeds per capsule (S/C) and the breeding system categories. Fully autogamous species had, on average, the highest number of seeds per capsule, whereas fully xenogamous taxa had the lowest (Fig 4). In the regression tree, the first split separated a leaf of 14% facultative autogamous taxa from the remainder of species, followed by additional splits based on their S/C averages (Fig 5). The terminal leaves of these additional splits divided the remaining 86% of taxa into additional leaves illustrating a steady increase of P/O values as the S/C average decreased (Fig 5).

## Seed size and distribution range of *Cuscuta* species in North America

Spearman's Rank correlation indicated a lack of correlation between the seed size and the total geographical distribution range of species (r = 0.0944, $r^2$ = 0.0089). This suggests that seed size does not have an impact on the dispersal and the total geographical range of the species in N America.

## Seedling survival

There was a significant difference in seed size among the three species studied (S5 Table). Kaplan-Meier seedling survival curves showed that seedlings originating from larger-sized seeds survived significantly longer than seedlings of smaller-sized seeds (Fig 6). The seedlings of *C. campestris*, having the largest seeds among the three species examined, survived longer (48.12 days) than the seedlings of *C. costaricensis* (36.79 days) and *C. epithymum*, respectively (20.49 days).

## Discussion

### Potential use of seed characters for systematics and identification of *Cuscuta* species

This is the most comprehensive study of *Cuscuta* seeds to date, examining for the first time 80 species and filling either morphological or anatomical knowledge gaps for the 21 previously

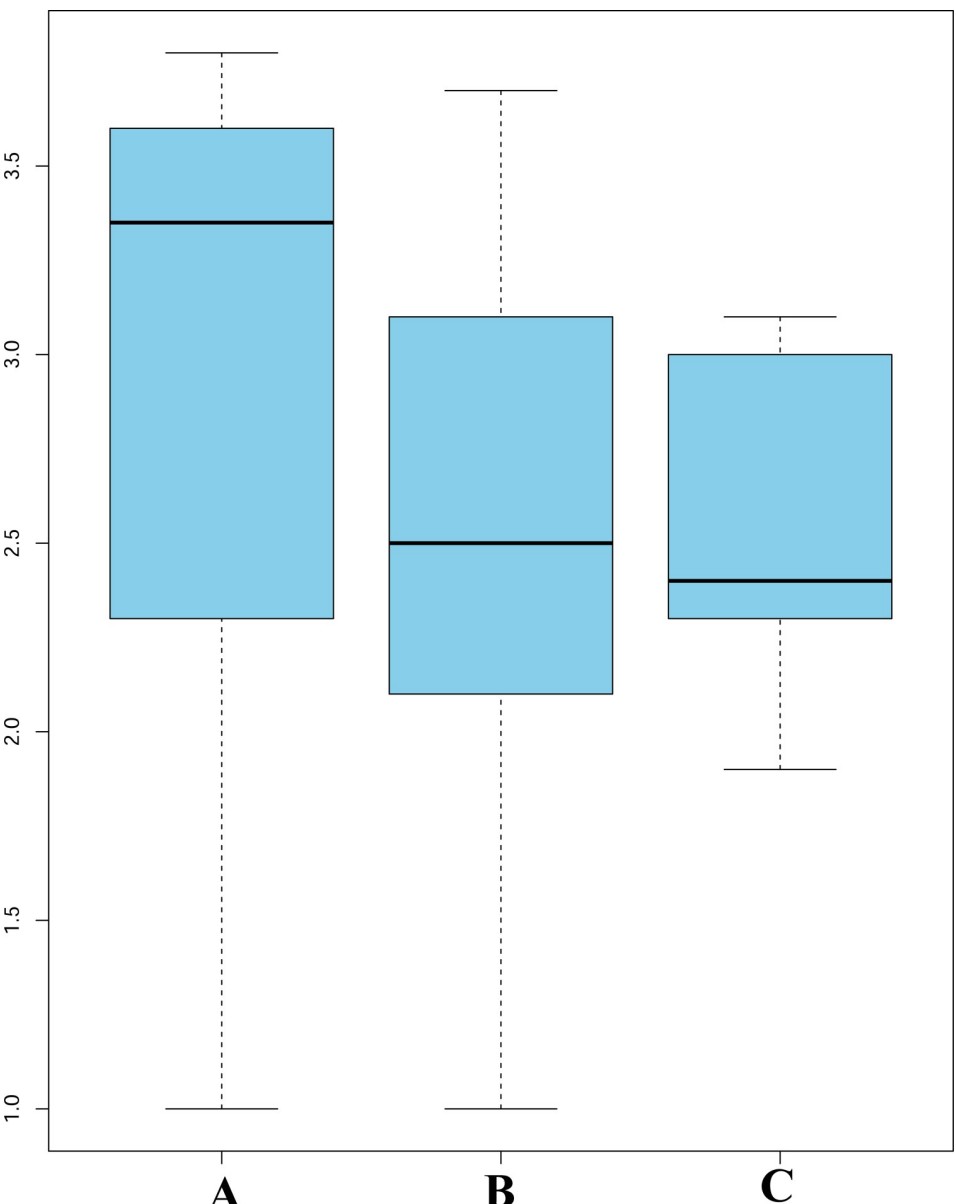

**Fig 4. Analysis of variance for the breeding system categories and the average number of seeds per capsule (S/C).**
A. Facultatively autogamous taxa have the highest S/C averages, but also the highest variation. B. Facultatively xenogamous group include species that possess an intermediate S/C number between the other two categories. C. Fully xenogamous taxa have the lowest S/C average and the least amount of variation; species in this category are self-incompatible.

studied species (S1 Table). Our findings largely confirmed previous studies (S1 Table), as well as earlier taxonomic revisions in which seeds were superficially described (mostly in terms of size and sometimes shape [e.g., 16, 17, 50, 51, 54–64]). Despite the significant increase of the taxon sampling and the fine level of morphological and anatomical detail employed, only a few additional features with systematic value were discovered. Species of subg. *Monogynella* are distinct compared to the rest of the genus because of their singular epidermal surface morphology, larger size, and single-palisade layer architecture of testa outside the hilum area. The remaining subgenera of *Cuscuta* can also be distinguished in most cases using a combination

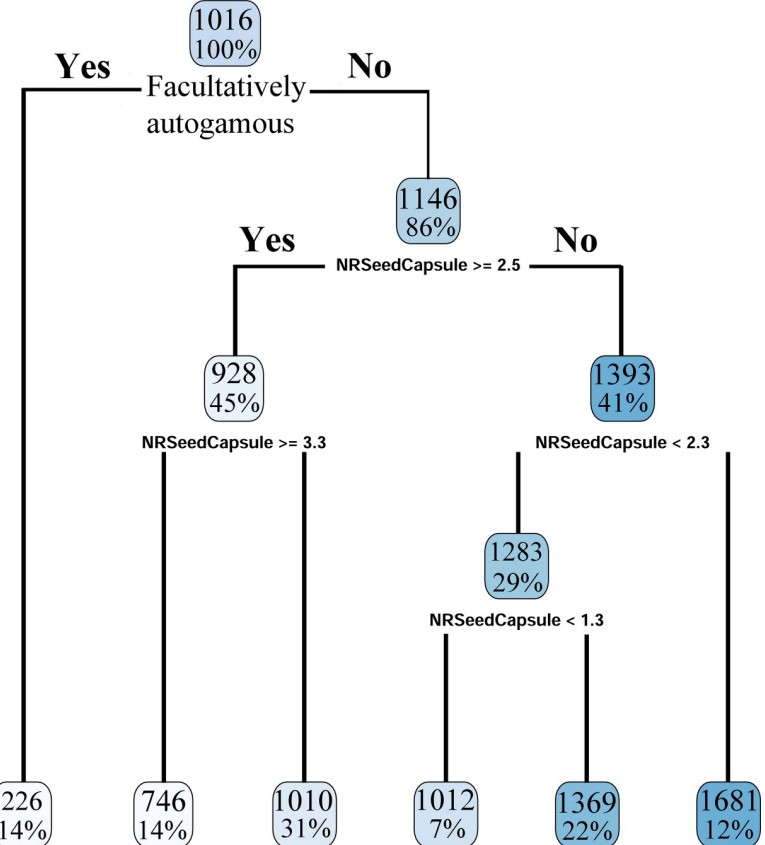

**Fig 5. Regression tree analysis of number of seeds per capsule (NRSeedCapsule) and pollen/ovule ratios (P/O) used as an indicator of breeding systems.** The first split separated directly the leaf of 14% facultatively autogamous taxa with an average P/O of 226 (first leaf to the left). At the next node, the remainder species were divided depending on whether they had more or less than 2.5 S/C. 45% of taxa had more than 2.5 S/C and were split again depending whether they had more or less than 3.3 S/C. 14% of taxa had more than 3.3 S/C and were placed in the second terminal leaf, with a P/O of 746. 31% had less than 3.3 S/C and were separated in the third terminal leaf, with a P/O of 1010. Taxa with more than 2.3 S/C were found in the sixth terminal leaf, comprising 12% of the total, P/O of 1681. Taxa with less than 2.3 S/C were divided once more if they have more of less than 1.3 S/C. 7% of the total had less than 1.3 S/C, P/O 1012, while 22% had more than 1.3 S/C and P/O of 1369.

of morphological and anatomical traits (S2 Table). The 15 sections of subg. *Grammica* [1], however, cannot be separated because of the high level of homoplasy observed (although some exceptions exist; e.g., sect. *Denticulatae*). Although not systematically significant at a sectional level, seed morphology and anatomy can provide valuable taxonomic data for identification purposes. As indicated by other authors (e.g., [9, 12, 20]), species identification is difficult by seed characters alone, but not impossible if the geographical origin of seeds is known which reduces the number of potential species from among which the identification starts. If the geographical origin of seeds is unknown, identification can be narrowed down to subgenus, and species recognition can be completed using a molecular approach using the sequences we have uploaded in Genbank for numerous species. In this latter case, we advise caution as many of the sequences uploaded in Genbank may have originated from misidentified plants (see the discussions in [51, 63]) and the systematics of many *Cuscuta* clades is still unresolved at a species level.

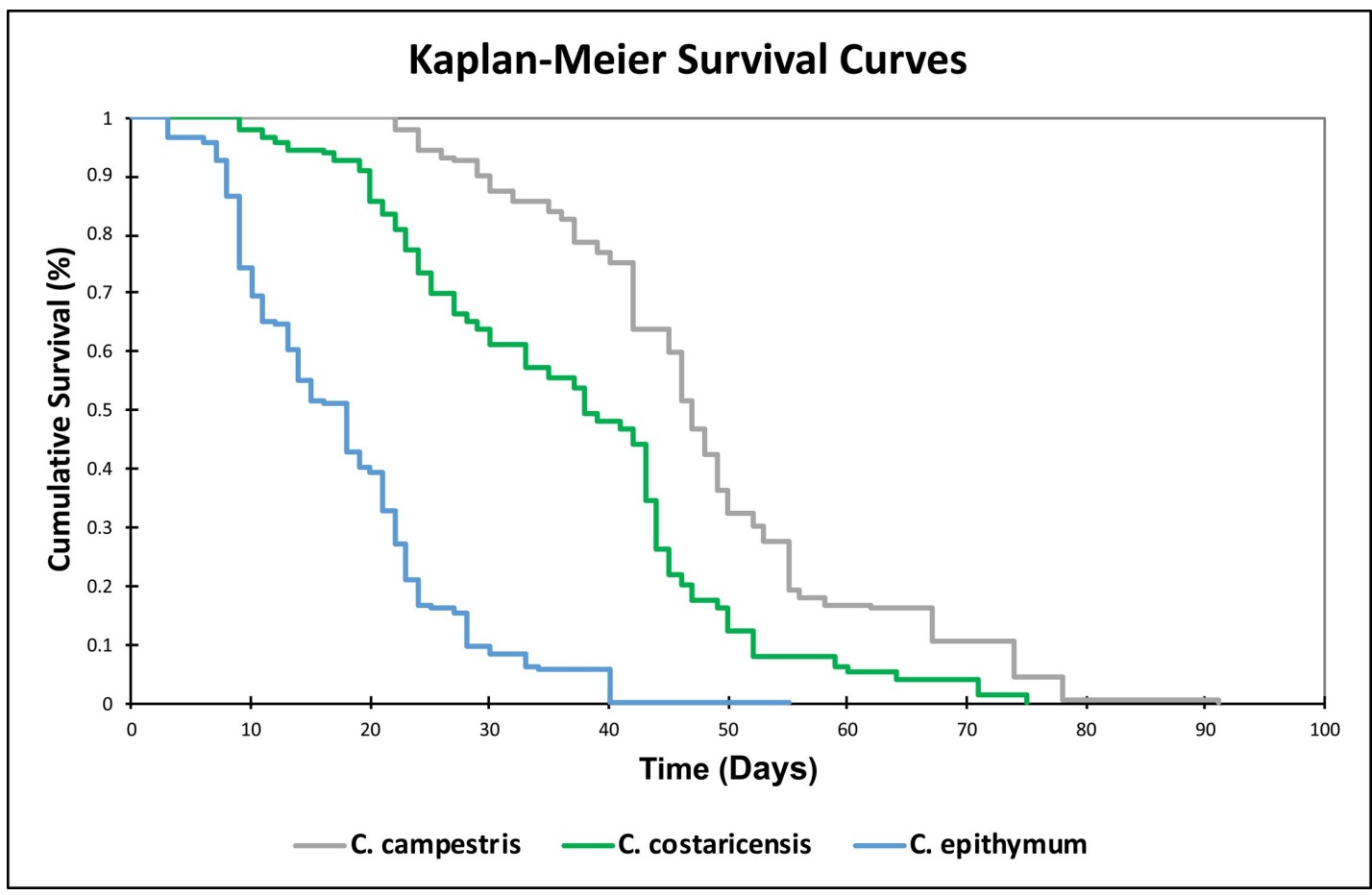

**Fig 6. Kaplan-Meier survival curves showing the proportion of *Cuscuta* seedlings surviving (log scale).** Species are listed in order of their increasing seed size: blue line = *C. epithymum*, the smallest seeds (0.89 mm in average); red line = *C. costaricensis*, intermediate sized-seeds (1.08 mm in average); green line = *C. campestris*, the largest seeds (1.28 mm in average). Standard error not shown for clarity.

## Evolution of form and function in *Cuscuta* seeds

Several studies [e.g., 21, 65] indicated that similarities exist between the surface of the seed coat in *Cuscuta* and other genera in Convolvulaceae (e.g., *Convolvulus*, *Cressa*, *Evolvulus*, *Ipomoea*, and *Seddera*). Despite the fact that the epidermal cells of some Convolvulaceae taxa are isodiametric and more or less dome-shaped, resembling those of *Cuscuta* (e.g., [66–68]), it is unknown if they can alternate from pitted to papillose depending on their hydration status. This interesting trait observed in all the *Cuscuta* species with the exception of subg. *Monogynella*, is apparently ensured by flexible nature of the outermost cell walls of the epidermal cells [12, 19, 69] and the presence of hydrophilic pectic zones, capable of attracting and retaining water [12, 20, 69].

Lyshede [20] and [70] suggested that the pitted epidermis morphology of *Cuscuta* dry seeds is an adaptation for wind dispersal. However, dodder seeds are much larger than typical anemochory adapted seeds like those of Orobanchaceae or some Plantaginaceae, which are "dust-like" and exhibit clearly reticulate or "honey-comb" epidermal morphologies [71–73]. Based on the results of this study, it is more likely that the pitted/papillate seed coat is an adaptation related to the imbibition, and more broadly, germination. The presence of pectin in the cuticle and epidermal cell walls results in the creation of a mucilaginous layer around the seeds when

seeds are hydrated [6, 20, 74], protecting them from desiccation [75] and allowing them to adhere to soil particles. This epidermis trait also allows seeds to be dispersed via farm machinery [6] and bird epizoochory (e.g., [76]). Last but not least, the hydrated epidermis prevents the displacement of seeds within the soil profile, which is important because *Cuscuta* species require light for germination [77–80]. These functional advantages of the pitted/papillose seed epidermis support the result of the likelihood ancestral reconstruction according to which this trait is derived from the morphologically invariant epidermis encountered in subg. *Monogynella.*

Corner [53] supported the classification of *Cuscuta* in Convolvulaceae based on the ontogeny and structure of the seed coat, which is shared by all the studied family members [67, 81, 82]. The inner palisade (or single palisade layer of subg. *Monogynella* and several *Grammica* species) contains a light line (*linea lucida*) similar to that present in the seeds of other taxa with impermeable seed coats (e.g., Convolvulaceae, Cucurbitaceae, Fabaceae, Geraniaceae, Malvaceae; reviewed by [83]). An impermeable palisade cell layer(s) or "hardseedness" has been associated with physical dormancy in many angiosperms, including in *Cuscuta* and *Convolvulaceae* [83–87]. In *Cuscuta*, the inner (or single palisade when only one palisade layer is present outside hilum area) is strongly lignified and it represents the ultimate mechanical defense of the seeds. Bird endozoochory experiments reported that gut passage partially or entirely stripped the outer palisade layer, while the inner or single palisade layer was little affected [13, 14]. If the function of the inner palisade layer (or single one in *Monogynella*) is to safeguard the impermeability of the seed coat, ensure physical dormancy and provide an ultimate line of mechanical defense, the role of the external palisade layer is not clear. The cells of the external palisade layer are in close contact with the epidermal cells, and in addition to protection, they may be involved in the opening of the hilar water gap.

Current findings showed that the outer palisade layer can be lost through evolution in the *Grammica* species with a globose embryo. All the above mentioned subtle adaptations for dispersal, e.g., via epi- or endozoochory, indicate that the characterization of *Cuscuta* seeds as "unspecialized" [e.g., 10–12] is not suitable (see also the discussion in [14]).

Originally, it was thought that the papillae on *Cuscuta* seed epidermis represented the access point of water to the embryo [20, 78]. Our results confirmed the findings of [22] in *C. australis*, in that the water enters into intact seeds through the hilar fissure/water gap. We found that the water gap anatomy in *Cuscuta* is uniform despite variations observed in the structure of the testa (e.g., epidermis type and number of palisade layers). In other Convolvulaceae seeds (e.g., *Ipomoea*, *Merremia*, *Calystegia*) "bulges" adjacent to the micropyle initiate the water entry into the seed [88, 89]. In *Cuscuta*, the opening of the hilar fissure during imbibition may be the result of the overall tensions created within the palisade layer(s) (particularly the external one) by the turgescent epidermal cells.

Many authors have mentioned or investigated the development of the filiform, coiled embryo in *Cuscuta* [12, 20, 78, 90–94]. The coiling of the embryo has been said to foreshadow the parasitic behavior of the plant as it coils around its host [10]. It is more conceivable, as [94] suggested, that the coiling pattern allows for a longer embryo to develop within the limited space of the seed, and upon germination, the extra length gained through the straightening of the coils may represent an advantage for the seedlings that need to elongate rapidly in order to locate a host. *Grammica* species of sect. *Denticulatae* and *C. microstyla* (sect. *Subulatae*) evolved an embryo with an enlarged spherical or club-shaped radicular end. A new species, *C. psorothamnensis* belonging to sect. *Denticulatae*, was recently described from Anza-Borrego desert in southern California [95]. This species was not examined in the current study, but it was reported by [95] to also possess an embryo with a globular swelling toward the radicular end. The species of sect. *Denticulatae* [55, 95] are viviparous: their seeds germinate while still

found inside the capsules and the parasite is attached to the host. This ensures that the seedlings will be able to attach directly to the shoots of same host plant. Considering the desert habit these species occur in [55, 95], it is most likely that this peculiar embryo has evolved as a storage organ. Less is known about the natural history of *C. microstyla* (section *Subulatae*), but this species also grows in arid habitats in the Andes [55, 96].

## Allometric relationships of seeds and seedling survival

We have showed that autogamous species had the highest number of seeds per capsule (S/C) whereas fully xenogamous taxa had the lowest. Having a mixed mating system, allows *Cuscuta* species to combine in different proportions the reproductive assurance of selfing with the boost of genetic diversity provided outcrossing [27, 97, 98]. High measures of reproductive output (e.g., seed/ovule ratio, number of seed/fruit) characterize annuals, while these measures are generally lower for perennials, which are more often outcrossing or clonal [e.g., 99–102]. *Cuscuta* species are usually considered to be annual [e.g., 3, 4, 8, 19]. However, many species growing on perennial herb hosts and especially woody hosts behave as perennial because they can regenerate yearly from haustorial tissue left inside the host [6, 8, 103–106]. We have often noted in the field in Mexico that *Cuscuta* species characterized as fully xenogamous by [27] (e.g., *C. volcanica*) are parasitic on woody plants and "perennial" from an endophyte, while facultatively autogamous dodders grow on annual hosts and synchronize their life cycle with them. This potential relationship has not been studied to date, but considering that most weedy and invasive dodders comport as annual [3, 4, 6], a possible connection between the seed production, breeding system and host range would be interesting to investigate in the future.

Seed size is strongly related some plants with their dispersal ability [107–109], but in *Cuscuta* we have found no indication of such a relationship. Using the same geographical dataset, [15] reported that distribution patters of subg. *Grammica* in N America are strongly associated with the dehiscence or indehiscence of capsules. As the current results suggested, it is more likely that seed size investment in *Cuscuta* is related to the seedling survival rather than with the dispersal capability.

In many other plants, seed size has been shown to be positively correlated with higher seedling survivorship rates when seedlings face unfavorable conditions (e.g., drought, deep shade, high depth burial within the soil) because larger seeds have more food reserves [e.g., 110–112]. In the case of *Cuscuta*, the seedling stage ontogenetic bottleneck is even more critical because in addition to surviving abiotic and biotic challenges similar to green plants [e.g., 35, 113–115], seedlings must also locate and overcome within a limited amount of time the defenses of compatible hosts [e.g., 4, 6]. Seed size affects seedling survival time because seedlings are unable to photosynthesize and thus they depend entirely on their seed reserves. The long survival times reported in this study, 40 to 90 days depending on the species, are very unlikely to be found under natural conditions because seedlings were fully protected against desiccation and "death" was noted only when seedlings were entirely necrotic. Survival times ranging from one to several weeks are more likely to be found under natural conditions as reviewed by [36].

## Supporting information

**S1 Fig. Parsimony ancestral reconstruction of number of embryo coils in *Cuscuta* seeds.** Embryos with more than 2.75 coils evolved multiple times in subg. *Grammica*. (TIF)

**S1 Table. *Cuscuta* species previously studied for seed morphology and/or anatomy arranged alphabetically and indicating their publication source.** "+" and "—" indicate presence or absence of data.
(DOCX)

**S2 Table. Seed character dataset for *Cuscuta*.** Refer to Table 1 for the character states. CP = Compression; S = Shape; ET = Embryo type; HP = Hilum position; HC = Hilum compression; Dep = Dry seed epidermis; Hep = Hydrated seed epidermis; ECS = Epidermal cell shape; OP = Presence of outer palisade layer; #C = number of embryo coils; L = Seed length (μm); W = Seed width (μm); ST = Seed thickness (μm); Hl = Hilum length (μm), HW = Hilum Width (μm); FL = Length of funicular scar (μm); ECD = Epidermal cell diameter (μm); EPT = Epidermal cell thickness (μm); Epidermal cell width (μm).
(DOCX)

**S3 Table. Summary of basic statistics for quantitative characters of *Cuscuta* seeds.**
(DOCX)

**S4 Table. Pearson's correlation summary for quantitative seed characters of *Cuscuta*.**
(DOCX)

**S5 Table. Seed size of three *Cuscuta* species studied and summary of statistical results.**
(DOCX)

**S1 Appendix.**
(DOCX)

# Acknowledgments

Catherine Swytink-Binnema and Hiba El Miari helped with the preparation of some of the SEM samples. We are grateful to the curators and herbarium managers who sent their *Cuscuta* herbarium specimens on loan to WLU. Ádám Lovas-Kiss kindly provided one of the samples of *C. lupuliformis*. Frédérique Guinel and Kevin Stevens were part of the MSc advisory committee of the first author and provided useful comments to the thesis, which served as a basis for this article. Derek Gray helped with advice for the logistical regression analysis. Last but not least, we thank to Alexander Sukhorukov and three anonymous reviewers for their suggestions.

# Author Contributions

**Conceptualization:** Mihai Costea.

**Data curation:** Magdalena Olszewski, Meghan Dilliott, Ignacio García-Ruiz, Behrang Bendarvandi.

**Formal analysis:** Magdalena Olszewski, Meghan Dilliott, Mihai Costea.

**Funding acquisition:** Mihai Costea.

**Investigation:** Magdalena Olszewski.

**Methodology:** Magdalena Olszewski, Mihai Costea.

**Supervision:** Mihai Costea.

**Visualization:** Magdalena Olszewski, Mihai Costea.

**Writing – original draft:** Magdalena Olszewski, Mihai Costea.

**Writing – review & editing:** Mihai Costea.

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
