## [Decision Letter · Decision Letter 0]

13 May 2020

PONE-D-20-10692

Cuscuta seeds: diversity and evolution, value for systematics/Identification and exploration of some allometric relationships

PLOS ONE

Dear Mihai Costea,

Thank you for submitting your manuscript to PLOS ONE. After careful consideration, we feel that it has merit but does not fully meet PLOS ONE’s publication criteria as it currently stands. Therefore, we invite you to submit a revised version of the manuscript that addresses the points raised during the review process.

The reviewers were enthusiastic about your study, as was I. However,  the reviewers brought up important issues that need to be addressed in a new revision. Thank you for conducting such an interesting study, and I look forward to reading your revision.

We would appreciate receiving your revised manuscript by 12 of July. To enhance the reproducibility of your results, we recommend that if applicable you deposit your laboratory protocols in protocols.io, where a protocol can be assigned its own identifier (DOI) such that it can be cited independently in the future. For instructions see: http://journals.plos.org/plosone/s/submission-guidelines#loc-laboratory-protocols

We look forward to receiving your revised manuscript.

Kind regards,

Guillermo C. Amico

Academic Editor

PLOS ONE

Additional Editor Comments:

The reviewers were enthusiastic about your study, as was I. However, the reviewers brought up important issues that need to be addressed in a new revision. Thank you for conducting such an interesting study, and I look forward to reading your revision.

Regards

Guillermo

2. We noted in your submission details that a portion of your manuscript may have been presented or published elsewhere:

'We used data from the following to articles:

1) Pollen/ovule ratios from Wright, M.A., Ianni, M.D. and Costea, M., 2012. Diversity and evolution of pollen-ovule production in Cuscuta (dodders, Convolvulaceae) in relation to floral morphology. Plant Systematics and Evolution, 298(2), pp.369-389.

2) Geographical area size for Cuscuta species in N America from:

Ho, A. and Costea, M., 2018. Diversity, evolution and taxonomic significance of fruit in Cuscuta (dodder, Convolvulaceae); the evolutionary advantages of indehiscence. Perspectives in Plant Ecology, Evolution and Systematics, 32, pp.1-17.'

Please clarify whether these publications were peer-reviewed and formally published. If this work was previously peer-reviewed and published, in the cover letter please provide the reason that this work does not constitute dual publication and should be included in the current manuscript.

Reviewers' comments:

Reviewer's Responses to Questions

**Comments to the Author**

1. Is the manuscript technically sound, and do the data support the conclusions?

Reviewer #1: Yes

Reviewer #2: Yes

Reviewer #3: Yes

Reviewer #4: Yes

2. Has the statistical analysis been performed appropriately and rigorously? 

Reviewer #1: Yes

Reviewer #2: Yes

Reviewer #3: Yes

Reviewer #4: Yes

3. Have the authors made all data underlying the findings in their manuscript fully available?

Reviewer #1: Yes

Reviewer #2: Yes

Reviewer #3: Yes

Reviewer #4: Yes

4. Is the manuscript presented in an intelligible fashion and written in standard English?

Reviewer #1: Yes

Reviewer #2: Yes

Reviewer #3: Yes

Reviewer #4: Yes

5. Review Comments to the Author

Reviewer #1: Dear all,

The paper is interesting and should be publushed in Plos One. I have attached a file with some queries and suggestions, and I hope it helps to improve some parts of the paper.

Best wishes,

Alexander

Reviewer #2: It is "the most comprehensive study" on Cuscuta species, seeds and their associated classification and function.

Only suggestion is to add more information on following:

1, please define the "length", "width" and "thickness" in Table 1 line 12-14, it will increase the clarity. e.g., the first, second or third longest axis for length, width and thickness.

2, on the line of 189-190 to add seed year for those three species if the data available. Or seed year collected are comparable for the survival study since the longevity may play a role in their survival time.

Reviewer #3: This work is the most comprehensive study to date on the seed morphology and anatomy of the genus Cuscuta. Sampling includes about half of the species of the genus and all the sections currently recognized except for the small sect. Epistigma. A detailed revision of seed morphology and anatomy is complemented with the study of embryo morphology, number of seeds/capsule related to breeding systems, seed size and distribution range in N America, germination and seedling survival. All of them are analyzed in a robust phylogenetic framework. It is a robust work that will be a reference for future studies of seed biology in this economically and ecologically important genus of obligate parasitic plants. The paper is well written and the information presented clearly merits its publication in PlosOne. Only a few minor corrections might help to improve an already excellent paper in its current state.

- Lines 357-358; 394. There are four (not three) species described in this section, including C. psorothamnensis, sharing the embryo enlarged toward the radicular end typical of the section.

- It is not clear the anatomical structure in the hilar area. Most species have a continuous double palisade layer except for subgenus Monogynella and four (five?) species on Grammica in which the outer palisade layer is only present in the hilar area. Therefore the anatomy in subgenus Monogynella is two palysade layers and the epidermis above the endosperm in the hilar area. Figure 3E-F shows a stained thickened structure with the tracheid-like line between the external surface of the hilum and the endosperm. I could not find and explanation of what structure is that. Is it part of the endosperm or a swelling of the palisade layers? There is a significant increase in thickness of the inner palisade layer in the hilum pad but up two times (line 363). Therefore it cannot be the inner palisade layer because it is much thicker than that. This is confusing and should be explained more clearly. It would be useful to point with arrows the two palysade layers of the hilar pad.

- What is the line of the inner palysade layer (P1) shown in figure 1-O? Compare with the photograph of C. pacifica in Fig. 2. Comparing both images it could be interpreted that what is labeled as P2 in Fig. 1-O is actually the epidermis. In Fig. 1-N, what is the structure between the endosperm (En) and P1?

-I might be wrong but I think there is a mistake in the legend of Fig. 5, lines 431-433. Taxa with more than 2.3 S/C were divided if they had "more of less" than 1.3 S/C. If the taxa had more than 2.3 S/C one of the groups cannot have less than 1.3 S/C. Add also a space in line 429 to separate 746 and 31 because it reads "746.31%".

Minor corrections:

-Sentence in lines 56-57. It is not clear what it means.

-Correct sentence in lines 208-209.

-Line 471, add parenthesis to the list of references.

Reviewer #4: Dear Authors,

The submitted manuscript seems to me a very important contribution to the knowledge of seed anatomy, morphology, and evolution and development in the genus Cuscuta, which is not only of great economic importance, but also very relevant for laying a foundation for further understanding the evolution and dispersal mechanisms of this parasitic plant, with impact on host-parasite interaction studies (ecology and evolution). It is a very comprehensive work, with a great quantity of data that must have required years of study, and analysed with appropriate methodological strategies, to the best of my knowledge. It will also be a good model for similar studies in the family Convolvulaceae, which are very much necessary.

Therefore, I am very supportive of the publication of these research results, for even with sampling limitations, it will leverage an array of other studies in the fields of plant systematics, ecology and evolution in this, and other, groups of plants.

My most significant recommendations go towards the actual presentation of the results, i.e. how the manuscript is written and structured. The importance of making these data and analyses available is undeniable, but I would strongly recommend to present it in a more effective way.

I have felt that the research hypothesis and how the applied methodology will address them are not laid out clearly enough, and in my view the contents are mixed up both in the Introduction, Methodology and Discussion. A more structured organization of the contents would allow summarizing some parts thar are overlapping between sections, and expanding on some discussions which would be interesting to extend.

As I understand it, this study is decomposed in the following components:

1) What is the variation? Documenting the variation in seed morphology and anatomy and coding this variation for further analysis; also, explicitly laying out the taxonomic and biogeographic scope of the study (if it is not the same for each section, I do not see a problem, but it needs to be explicitly defined from the outset).

2) How has this variation evolved in the group? Testing character evolutionary hypothesis in a molecular phylogenetic framework

3) Can this variation contribute to the systematics of the group? Given the character evolution, can synapomorphies be found for particular clades, and are they useful or not for diagnosing taxa of any particular rank?

4) Character correlations – do certain traits correlate to each other, and possibly why?

5) Additional information about seed development

The clear structure of contents would allow greater transparency and clarity in the communication of the hypotheses, and interpretation of the results. For instance, I understand that the characters explored did not contribute in a meaningful way to the systematics of the group. However, this is a result in itself, and further corroborates the already documented homoplasy present in so many characters in family Convolvulaceae. This needs to be openly addressed and discussed, as the other research questions.

My suggestion for a re-structuring of the manuscript is to

- offer the objectives in the form of research questions or scientific problems (in the way they are presented, they seem more like a narrative of the methodology)

- Present the results and discussion as answers to these research questions, therefore following a particular order of contents within each section, e.g. documentation of the variation, evolutionary analyses, value of characters for systematics and correlations between characters

For further details, I have added comments throughout the manuscript draft, but several of those will relate to this general organization of contents that I have laid out, and which I think hinders the value of this publication as is.

In summary, I recommend the manuscript to be accepted with major revisions, and I would be happy to revise an improved version.

Kind regards,

6. PLOS authors have the option to publish the peer review history of their article (what does this mean?). If published, this will include your full peer review and any attached files.

Reviewer #1: Yes: Alexander P. Sukhorukov

Reviewer #2: No

Reviewer #3: No

Reviewer #4: No

---

## [Author Response · Author response to Decision Letter 0]

22 May 2020

Please see the two word documents attached (response to reviewers #1 and #2). The responses are likely to extensive for appropriate view here.

---

## [Decision Letter · Decision Letter 1]

1 Jun 2020

Cuscuta seeds: diversity and evolution, value for systematics/Identification and exploration of allometric relationships

PONE-D-20-10692R1

Dear Dr. Mihai Costea,

We are pleased to inform you that your manuscript has been judged scientifically suitable for publication and will be formally accepted for publication once it complies with all outstanding technical requirements.

With kind regards,

Guillermo C. Amico

Academic Editor

PLOS ONE

Additional Editor Comments (optional):

The reviewers coincide that comments have been addressed and did not find any further concerns. The manuscript will have a substantial impact on this field, congratulations!

Reviewers' comments:

Reviewer's Responses to Questions

**Comments to the Author**

1. If the authors have adequately addressed your comments raised in a previous round of review and you feel that this manuscript is now acceptable for publication, you may indicate that here to bypass the “Comments to the Author” section, enter your conflict of interest statement in the “Confidential to Editor” section, and submit your "Accept" recommendation.

Reviewer #2: All comments have been addressed

Reviewer #3: All comments have been addressed

2. Is the manuscript technically sound, and do the data support the conclusions?

Reviewer #2: Yes

Reviewer #3: Yes

3. Has the statistical analysis been performed appropriately and rigorously? 

Reviewer #2: Yes

Reviewer #3: Yes

4. Have the authors made all data underlying the findings in their manuscript fully available?

Reviewer #2: Yes

Reviewer #3: Yes

5. Is the manuscript presented in an intelligible fashion and written in standard English?

Reviewer #2: Yes

Reviewer #3: Yes

6. Review Comments to the Author

Reviewer #2: Thanks for addressing all the comments. It is a comprehensive study with sound research methodology and approaches. Support the publication in full. Congratulations to the good work!

Reviewer #3: The authors have addressed all my concerns from the first version of the manuscript:

-The 4th species in sect. Denticulatae is now mentioned in the discussion.

-Now it is clear that the thick structure in the hilar area is part of the water gap and is composed of parenchyma between the two palisade layers and the endosperm. The new figure 3F clearly illustrates the two palisade layers in the hilar area.

-Legend and figure 5 have been corrected. The legend still has written "7%" in bold font and should be corrected.

-Labels in Fig. 1O have been fixed.

-Other minor corrections have been addressed.

Briefly, this manuscript is ready for its publication after resolving possible issues by other reviewers. In my opinion it is clearly written, the results and discussion correctly structured. Of course, a work like this, with so many analyses and so much information could be presented in many different ways. But I think that the authors did a great job presenting the objectives of the work, the analyses performed, the results obtained and the discussion of the results in an evolutionary framework as well as its systematics implications. Congratulations.

7. PLOS authors have the option to publish the peer review history of their article (what does this mean?). If published, this will include your full peer review and any attached files.

Reviewer #2: No

Reviewer #3: No

---

## [Editor Report · Acceptance letter]

3 Jun 2020

PONE-D-20-10692R1 

Cuscuta seeds: diversity and evolution, value for systematics/Identification and exploration of allometric relationships 

Dear Dr. Costea:

I'm pleased to inform you that your manuscript has been deemed suitable for publication in PLOS ONE. Congratulations! Your manuscript is now with our production department. 

Kind regards, 

on behalf of

Dr. Guillermo C. Amico 

Academic Editor

PLOS ONE